# Palm Tocotrienol Activates the Wnt3a/β-Catenin Signaling Pathway, Protecting MC3T3-E1 Osteoblasts from Cellular Damage Caused by Dexamethasone and Promoting Bone Formation

**DOI:** 10.3390/biomedicines13010243

**Published:** 2025-01-20

**Authors:** Norfarahin Abdullah Sani, Nur Aqilah Kamaruddin, Ima Nirwana Soelaiman, Kok-Lun Pang, Kok-Yong Chin, Elvy Suhana Mohd Ramli

**Affiliations:** 1Department of Anatomy, Faculty of Medicine, Universiti Kebangsaan Malaysia, Cheras, Kuala Lumpur 56000, Malaysia; drfarahin.as@gmail.com (N.A.S.); nur.aqilah@ukm.edu.my (N.A.K.); 2Department of Pharmacology, Faculty of Medicine, Universiti Kebangsaan Malaysia, Cheras, Kuala Lumpur 56000, Malaysia; imasoel@ppukm.ukm.edu.my (I.N.S.); koklun.pang@monash.edu (K.-L.P.); chinkokyong@ppukm.ukm.edu.my (K.-Y.C.); 3Jeffrey Cheah School of Medicine and Health Sciences, Monash University Malaysia, Bandar Sunway, Subang Jaya 46150, Malaysia

**Keywords:** bone, differentiation, osteoporosis, tocotrienol, vitamin E, glucocorticoids

## Abstract

**Background and aim:** Prolonged glucocorticoid (GC) treatment increases oxidative stress, triggers apoptosis of osteoblasts, and contributes to osteoporosis. Tocotrienol, as an antioxidant, could protect the osteoblasts and preserve bone quality under glucocorticoid treatment. From this study, we aimed to determine the effects of tocotrienol on MC3T3-E1 murine pre-osteoblastic cells treated with GC. **Methods:** MC3T3-E1 cells were exposed to dexamethasone (150 µM), with or without palm tocotrienol (PTT; 0.25, 0.5, and 1 µg/mL). Cell viability was measured by the MTS assay. Alizarin Red staining was performed to detect calcium deposits. Cellular alkaline phosphatase activity was measured to evaluate osteogenic activity. The expression of osteoblastic differentiation markers was measured by an enzyme-linked immunoassay. **Results**: Enhanced matrix mineralization was observed in the cells treated with 0.5 µg/mL PTT, especially on day 18 (*p* < 0.05). The expression of Wnt3a, β-catenin, collagen 1α1, alkaline phosphatase, osteocalcin, low-density lipoprotein receptor-related protein 6, and runt-related transcription factor-2 were significantly increased in the PTT-treated groups compared to the vehicle control group, especially at 0.5 µg/mL of PTT (*p* < 0.05) and on day 6 of treatment. **Conclusions**: PTT maintains the osteogenic activity of the dexamethasone-treated osteoblasts by promoting their differentiation.

## 1. Introduction

Because of its effects on both individuals and nations, osteoporosis has been ranked as the second most important health concern in the world. The economic toll that osteoporosis has on nations worldwide is substantial. The main cost drivers in this study were fracture-related costs [1]. A dynamic metabolic process that includes both bone production and resorption has a comparatively high turnover in bone. Bone formation largely depends on the proliferation and differentiation of osteoblasts to maintain strong bone structure and metabolic balance of calcium and phosphate in the body [2,3]. Any disruption to this process leads to osteoporosis or bone sclerosis. Secondary osteoporosis can be caused by a variety of causes, such as diabetes, genetics, endocrine diseases, malnutrition, and medication use [4,5]. Glucocorticoids (GCs) are steroid hormones that are produced or found naturally and have been used extensively to treat a variety of diseases due to their potent anti-inflammatory qualities [6]. However, long-term and excessive glucocorticoid use alters the bioactivity of osteoblasts/osteocytes, osteoclasts, and bone marrow-derived stem cells, leading to glucocorticoid-induced osteoporosis (GIO), the most common kind of secondary osteoporosis [7,8,9]. GCs reduce the viability and development of osteoblasts, which inhibits the formation of new bone [3,10]. The direct effects of GCs on bone formation are mostly linked to the upregulation of peroxisome proliferator-activated receptor gamma 2 (PPARγ2) and impact the Wnt/β-catenin signaling pathway [9,10]. Runt-related transcription factor 2 (Runx2) and Osterix (SP7) transcription factors are expressed in the presence of Wnt and BMP [11,12,13]. Previous studies have shown that GCs inhibit bone marrow-derived stem cell osteogenesis and inhibit Runt-related transcription factor 2 (Runx2) during mesenchymal cell osteoblast differentiation [8,14]. In addition, GCs have an impact on the mechanisms involving different bone formation pathways, including BMP, AKT, Mevalonat, and Wnt/β-catenin. Wnt, Receptor Activator of Nuclear Factor Kappa-B Ligand (RANKL), Osteoprotegerin (OPG), Macrophage-Colony Stimulating Factor (M-CSF), Follistatin, Dan, Dickkopf-1 (Dkk1), Wnt Inhibitory Factor 1 (WIF1), Sost, Secreted Frizzled-Related Protein 1 (sFRP-1), and Axin-2 are among the genes and biochemical expressions that are impacted by GCs. Transforming Growth Factor-β (TGF-β), Osterix (OSX), and Runt-Related Transcription Factor 2 (Runx2) [2,15,16]. Most studies were carried out to search for the prevention and treatment of GIO by targeting the regulation of bone biochemistry and remodeling pathways. A mutant mouse lacking or hypomorphic in Wnt3a was utilized in the first study to demonstrate the control and function of Wnt signaling on skeletal development and homeostasis [17,18]. Because it affects mesenchymal progenitors’ loyalty to the osteoblast lineage and the osteoblasts’ anabolic capacity to deposit bone matrix, Wnt/β-catenin signaling is crucial for achieving maximal bone mass. Wnt/β-catenin signaling is necessary for the initial stage determination of cells committed to the osteoblast lineage [19,20], and it also influences mature osteoblast performance [21,22,23], osteoclastogenesis [24,25] and osteoblast responsiveness to anabolic hormones [26,27,28,29].

The use of natural products in health care has been a subject of interest in disease treatments and prevention in recent years [30,31,32]. Since they are readily available, less expensive, and have been shown to have a variety of biological effects as well as improve bone quality, they provide an endless source of chemical diversity for the identification of new pharmacological modules [33,34]. One of the fat-soluble vitamins, vitamin E, is made up of two groups: tocopherol and tocotrienol. They have side chains and chromanol rings. While tocotrienol contains an unsaturated-sided chain with three double bonds that may help neutralize free radicals, tocopherol has a saturated-sided chain [35]. Numerous in vivo and in vitro investigations have demonstrated the potential of tocotrienol and tocopherol to maintain bones that experience osteoporosis alterations [29,34]. However, it appears that tocotrienol preserves bone cells better than tocopherol [36]. Tocotrienol’s ability to trigger the mechanisms that reduce GC-induced osteoporosis and glucocorticoid-induced osteoblast hyperstimulus requires more investigation. Liu et al. (2018) demonstrated that gastrodin (GSTD), a naturally occurring bioactive molecule with antioxidant properties, effectively increased osteogenic differentiation in MC3T3-E1 cells by modulating the NRF2 pathway [37].

Palm tocotrienols are tocotrienols derived from palm oil, which is a naturally occurring source of vitamin E. Palm oil contains a lot of both tocopherols and tocotrienols; however, the latter are more beneficial and bioactive. Tocopherol (18–22%) and tocotrienols (78–82%) make up palm vitamin E, which is found in the tocotrienol-rich fraction (TRF) of palm oil [38]. Numerous studies have been conducted on the potential health advantages of tocotrienols generated from palm oil, including its anti-inflammatory, neuroprotective, and antioxidant properties.

In this study, we investigated the effects of PTT on cell shape, proliferation, and differentiation using pre-osteoblastic MC3T3-E1 cells that had undergone osteogenesis suppression as a result of dexamethasone (DEX), a synthetic corticosteroid frequently used to treat inflammatory illnesses. PTT is thought to prevent a decrease in osteogenic activity in these cells via regulating Wnt/β-catenin signaling. Finding out if PTT, a natural material, might be utilized as a therapeutic to mitigate the adverse effects of high-dose DEX on MC3T3-E1 osteoblasts was the goal of this investigation. We plan to build on the findings of previous studies and develop PTT as a potential preventive medication that promotes bone formation in order to prevent osteoporosis caused by long-term GC treatment.

## 2. Materials and Methods

### 2.1. Chemicals

Alpha-modified essential media (AMEM), fetal bovine serum (FBS), and 0. 25%Tripsin-EDTA were purchased from GIBCO, Thermofisher, Watham, MA, USA. Penicillin–streptomycin and alizarin red dye were from Beijing Solarbio Science & Technology, Beijing, China and Sigma-Aldrich, St Louis, MO, USA. The Cell Titre 96 AQueous One Solution Cell Proliferation Assay (MTS) was purchased from Promega, Madison, WI, USA. The ALP assay kit was from Elabscience, Houston, TX, USA. Mouse LRP6, Wnt3a, CTNNb1, and Runx2 ELISA kits were purchased from Fine Test, Wuhan, China. Mouse COL1a1 and Osteocalcin ELISA kits were from Cloud-Clone Corp., Houston, TX, USA. All other compounds were analytical.

### 2.2. Cell Culture

The sample used was a pre-osteoblast Murine calvaria pre-osteoblast cell line MC3T3-E1 sub-clone 14 obtained from the American Type Culture Collection (ATCC), catalog number CRL-2594 (Manassas, VA, USA). The cells were grown in α-modified essential medium (α-MEM) (Thermo Fisher Scientific, Waltham, MA, USA) supplemented with 10% fetal bovine serum (Thermo Fisher Scientific) and 10% antibiotic–antifungal agent (Thermo Fisher Scientific) in humidified conditions at 37 °C and 5% carbon dioxide. In all experiments, the cells were subcultured at a density of 1 × 10^4^ cells/mL growth media. On the following day, the medium was changed into osteoblast differentiation medium (DM) with α-MEM supplemented with 3 mM sodium phosphate (Sigma-Aldrich Co., St Louis, MO, USA) and 50 µg/mL ascorbic acid (Sigma-Aldrich Co.). Ethanol was obtained from HmbG Chemicals (Hamburg, Germany). Lovastatin was obtained from ChemFaces (Wuhan, China). Passages 11–20 were used in all the experiments to maintain cell morphology and biological properties. During the 24 days of treatment, the medium was changed every 3 days.

Palm oil derived-tocotrienol (PTT) was a gift from Excelvite (Chemor, Perak, Malaysia), which contained 21.9% α-tocopherol, 24.7% α-tocotrienol, 4.5% β-tocotrienol, 36.9% γ-tocotrienol, and 12.0% δ-tocotrienol. PTT was prepared based on a previous study with some adjustments [34]. A total of 45 μL of 1 g/mL palm tocotrienol stock was activated by mixing it with 60 μL of FBS (GIBCO, Thermofisher, Watham, MA, USA) and kept in an incubator overnight at a temperature of 37 °C. The next day, it was diluted to 150 mg/mL by mixing it with 90 μL DM and 105 μL 99.8% pure ethanol (Chemiz, Shah Alam, Malaysia). Then, 300 μL of DM was mixed with 300 μL of pure ethanol (Chemiz, Shah Alam, Malaysia), and this solution was mixed into a 150 mg/mL solution to dilute it to 50 mg/mL before being used in the next dilution series: 0.05, 0.1, and 1 µg/mL. These dilutions were made fresh every two to three days until the experiment’s conclusion. The vehicle group’s treatment included the same quantity of ethanol as the PTT groups. The PTT was given daily together with DEX.

### 2.3. To Determine the Effects of Dexamethasone on Cell Proliferation and Differentiation

#### 2.3.1. Cell Viability Assay

The CellTiter 96 Aqueous One Solution Cell Proliferation Assay kit (Promega, Madison, WI, USA) was used to conduct the MTS assay. In summary, 96-well plates were seeded with MC3T3-E1 cells at a density of 1 × 10^4^ cells/well, and the cells were then cultured in the growth mixture for 24 h at 37 °C. The cells were then subjected to various DEX concentrations. Subsequently, the cells were exposed to different concentrations of DEX (0, 50, 100, 150, 200, 250, and 300 µM) for 3, 6, and 9 days. The treatment of DEX was administrated on a daily basis. Subsequently, 20 μL of MTS was added to each well, and incubation was continued for 4 h in a 37 °C incubator with 5% carbon dioxide (CO_2_). Light absorbance (OD) was recorded using an ELISA Multiscan 60 light absorbance reader (Thermofisher, Watham, MA, USA) at 490 nm. Each set of experiments was performed in triplicate. The cell viability was expressed as a percentage.

#### 2.3.2. Cell Morphology

MC3T3-E1 cells underwent morphological alterations both with and without DEX during mineralization and differentiation. A six-well plate containing plated MC3T3-E1 pre-osteoblast cells was cultured for one night. After that, the cells were given varying concentrations of DEX in differentiation media (DM) for three, six, nine, twelve, and twenty-four days. The EVOS Cell Imaging System (Thermo Fisher Scientific) inverted microscope was used to visualize the morphology of the cells after each time point. The cells were cleaned with phosphate-buffered saline (PBS), pH 7.4. Then, the cells were dissolved in 10% acetic acid and shaken overnight at room temperature. The absorbance was measured at 450 nm using a Multiskan Go microplate spectrophotometer (Thermofisher, USA).

### 2.4. To Determine the Effect of PTT on the MC3T3-E1 Cells Treated with Dexamethasone

A total of 45 μL of 1 g/mL palm tocotrienol stock was incubated by mixing it with 60 μL of FBS (GIBCO, Thermofisher, Watham, MA, USA) and kept in an incubator overnight at a temperature of 37 °C. The next day, it was diluted to 150 mg/mL by mixing it with 90 μL DM and 105 μL 99.8% pure ethanol (Chemiz, Shah Alam, Malaysia). Then, 300 μL of DM was mixed with 300 μL of pure ethanol (Chemiz, Shah Alam, Malaysia), and this solution was mixed into a 150 mg/mL solution to dilute it to 50 mg/mL before being used in the next dilution series according to the type of study.

#### 2.4.1. Alkaline Phosphatase (ALP) Assay

The early marker of bone formation was identified by measuring ALP activity. The plated cells were grown for the entire night on a six-well plate. The cells were next treated with DEX (150 M) and PTT (0.25, 0.5, and 1 µg/mL) for 6, 12, 18, and 24 days. At the end of each time point, the cells were harvested in accordance with the prescribed protocol, and the ALP assay kit (fluorometric) (Elabscience, US) was utilized to measure ALP activity.

#### 2.4.2. Mineralization Analysis by Alizarin Red Staining

To evaluate the impact of PTT on mineralization, the matrix was stained with Alizarin Red dye (Sigma-Aldrich Co.), which binds to calcium in the ECM, from day 6 to day 24. The cells were plated in a 48-well plate and allowed to incubate for the entire night. For 6, 12, 18, and 24 days, the cells were exposed to DEX (150 µM) and PTT (0.25, 0.5, and 1 µg/mL). Following each time point, the cells were fixed for ten minutes with 10% buffered formalin after being cleaned with PBS. After that, the wells were filled with 40 mM Alizarin Red (pH 4.4) and allowed to sit at room temperature for one hour. After being cleaned with PBS, the cells were examined using the EVOS Cell Imaging System, an inverted microscope with a ×100 magnification. Then, the cells were dissolved in 10% acetic acid and shaken overnight at room temperature. The absorbance was measured at 450 nm using a Multiskan Go microplate spectrophotometer (Thermofisher, Watham, MA, USA).

#### 2.4.3. Wnt3a, β-Catenin, COL1α1, and ALP and OCN, LRP6, and RUNX2 ELISA ASSAYS

After exposure to DEX (150 uM) and various concentrations of palm tocotrienol (0.25, 0.5, and 1 μg/mL) for 6, 12, 18, and 24 days, the activity of Wnt3a, β-catenin, COL1α1, and ALP and OCN, LRP6, and Runx2 were evaluated by ELISA kit, using the supernatant and the procedures were undertaken as directed by the manufacturer’s instructions. The absorbance was measured at 450 nm with a microplate reader (Thermofisher, Watham, MA, USA). All samples were examined in triplicate.

### 2.5. Statistical Analysis

All data are presented as means ± SEM of three independent experiments carried out in triplicates. Statistical analysis was performed using one-way analysis of variance (ANOVA) with Tukey’s post hoc test in IBM SPSS statistics version 22.0. A *p*-value of less than 0.05 (*p* < 0.05) was regarded as statistically significant.

## 3. Results

### 3.1. Effects of Dexamethasone on Cell Proliferation

This was to ascertain how dexamethasone affects cell division and evaluate its detrimental effects on MC3T3-E1 cells. Dexamethasone’s toxic effect started on day 3 when the treated cells showed a reduction in proliferation. However, it did not reach a significant value. The toxic effect of dexamethasone was more evident on day 6, where the cells treated with 200 μM, 250 μM, and 300 μM dexamethasone showed a significant reduction in the proliferation compared to the control group (*p* < 0.05) and lower dose treatment groups (10, 50 μM) (*p* < 0.05). (Figure 1a). As the treatment of dexamethasone continued until 9 days, the cell began to deform into an elliptical shape. The nuclei were visible, but the organelles became smaller and fainter. There was a decrease in the number of cells, and there was a widening of the space between the cells. The effects were worse in the 300 μM group, where fragments of dead cells were visible. IC25 was recorded at a concentration of 125 on day 9, while IC50 was recorded at a concentration of 285 μM on day 9 and 250 μM of dexamethasone on day 6 (Figure 1b).

### 3.2. Effects of DEX on the Morphology of MC3T3-E1 Cell Differentiation

Cells that proliferated then proceeded to differentiation and mineralization phases developed into mature cells. During mineralization, there were various biochemicals and minerals produced, among which were calcium nodules. Calcium nodules produced during cell ingestion were dyed with ARS and appeared as red lumps.

Each series of dexamethasone concentrations produced calcium nodules as early as the 12th day. On the 18th day, more and more nodulous calcium was produced, and the highest concentration of ARS red color was achieved on the 24th day. Higher concentrations of dexamethasone led to a lower production of calcium nodules.

On day 6 only, 200 and 300 μM groups showed a significant reduction in the calcium nodule production compared to control (*p* < 0.05). On day 12, the 150 μM group also showed a significant reduction in the calcium nodules. By day 18, there was a significant reduction in calcium nodules across all treatment groups as compared to the control group (*p* > 0.05). The calcium nodules significantly decreased as a result of dexamethasone treatment in a way that was dose- and time-dependent (*p* < 0.05) (Figure 2a,b).

### 3.3. PPT Reduced the Alkaline Phosphatase (ALP) Inhibitory Effect of DEX

The ALP enzyme produced by the dexamethasone-treated MC3T3-E1 cells supplemented with 0.5 μg/mL palm tocotrienol (D150T0.5), as well as 1 μg/mL of palm tocotrienol (D150T1), was significantly higher compared to the control (D150) on day 6 of the treatment. However, the enzyme level did not show any significant difference on days 12 and 18. However, on day 24, there was a significant reduction in the ALP level in the treatment groups compared to the control group, where the highest PTT group showed the lowest level. (Figure 3).

### 3.4. PPT Increased the Protein Level of LRP6, Wnt3a, CTNNb1, RUNX2, COL1a1 and OCN in DEX-Treated MC3T3-E1 Cells

#### 3.4.1. Protein LRP6

On day 6, cells treated with 0.5 μg/mL of palm tocotrienol (D150T0.5) recorded significantly higher LRP6 readings compared to the control group (D150). No significant changes were shown by the other two groups. There were no significant changes seen in any group on days 12 and 18. On day 24, the D150T0.5 group recorded significantly lower LRP6 levels compared to the control group (Figure 4).

#### 3.4.2. Protein Wnt3a

Cells treated with 0.5 μg/mL of palm tocotrienol (D150T0.5) showed significantly higher Wnt3a readings compared to control on day 6. There were no positive findings shown on the rest of the days (days 12, 18, and 24) (Figure 5).

#### 3.4.3. Protein CTNNb1

There was a significant increase in the CTNNb1 levels seen in the cells treated with 0.5 μg/mL (D150T0.25) and 0.5 μg/mL (D150T0.5) of palm tocotrienol on day 6. There were no significant positive results shown on the rest of the days (Figure 6).

#### 3.4.4. Protein RUNX2

The Runx2 expression was significantly increased in cells treated with 0.25 μg/mL (D150T0.25) and 0.5 μg/mL tocotrienol on day 6. Similar to earlier proteins, there were no significant positive results shown on the rest of the days (Figure 7).

#### 3.4.5. Protein COL1a1

On day 6, dexamethasone-treated cells given 0.25 μg/mL palm tocotrienol (D150T0.5) and 0.5 μg/mL palm tocotrienol (D150T0.5) expressed significantly higher COL1a1 compared to the control group (D150) (*p* < 0.05). No significant difference was seen in the D150T1 group. There were no positive changes seen on days 9, 18, and 24 (Figure 8).

#### 3.4.6. Protein OCN

Dexamethasone-treated cells added with palm tocotrienol 0.5 μg/mL (D150T0.5) and 1 μg/mL (D150T1) expressed significantly higher OCN compared to the control group (*p* < 0.05). The were no significant differences seen in OCN expression in any groups on days 12, 18, and 24 (Figure 9).

### 3.5. PPT Attenuated the Inhibitory Effect of DEX on Mineralization

Cell mineralization was quantified by measuring the deposition of calcium nodules. ARS staining method was used, and calcium nodules will appear as red spots. Calcium nodule was found to be significantly higher in the cells treated with 0.5 μg/mL palm tocotrienol (D150T0.5) compared to the control (D150) on day 18, with no significant changes seen in any other groups and doses (Figure 10a,b).

## 4. Discussion

Osteoporosis is brought on by an excess of glucocorticoid therapy because it reduces osteoblast viability and function [39]. High glucocorticoids level also leads to adipogenesis, causing fatty infiltration in the bone marrow, and in prolonged conditions, will also lead to hyperplasia of the adipocytes, which worsens the bone structure [40,41]. In this research, the impact of DEX on osteoblast viability was examined using MC3T3-E1 cells. In the present study, the establishment of an osteoporosis model induced by DEX treatment was successfully indicated by significantly decreased cell viability, osteoblastic differentiation, and reduction in the calcium nodules in a dose- and time-dependent manner in mouse MC3T3-E1 pre-osteoblasts. It has been shown that DEX inhibits the production of collagen and fibronectin while stimulating the synthesis of collagenase, which reduces the development of new bone [42,43,44,45]. According to earlier research, DEX dramatically reduced the viability of MC3T3-E1 cells at a concentration of 1 μM. It also appeared to limit the viability of MC3T3-E1 cells in a concentration-dependent manner. Additional research revealed that 1 μM DEX also resulted in G0/G1 phase arrest and apoptosis [46,47,48]. Chu et al. 2003 found that DEX causes MC3T3-E1 cells to undergo apoptosis by upregulating caspases-1, -3, -6, -8, -9, -11, -12 in MC3T3-E1 cells [49]. A study by Liu et al. 2018 indicated that DEX treatment of MC3T3-E1 significantly decreased cell viability, ALP activity, differentiation, and mineralization. There was also a reduction in the expression of Runx2, osterix, BMP-2, and osteocalcin (OCN) genes [45].

Osteoblasts release proteins that form the mineralization and bone matrix, and they are essential to the production of new bone. Agents that operate directly by either enhancing osteoblast proliferation or encouraging osteoblast differentiation are necessary to improve bone formation, as DEX-induced bone loss is predominantly mediated by osteoblast dysfunction. Numerous natural medicines with varying effects that show improvement in bone production, osteogenic differentiation, and skeletal repair have been studied [34,50]. PTT is rich in bioactive substances with significant medicinal and biological potential. It has also been shown to preserve bone in a number of animal osteoporosis models [51]. In our previous studies, we found that oral administration of PTT conferred beneficial effects in preserving the bones in glucocorticoid-induced osteoporosis in animal models [50]. Tocotrienol protects the bone against the detrimental effect of GCs by regulating the release of inflammatory mediators and reactive oxygen species (ROS). Nevertheless, more research is needed to determine how PTT affects osteoblastic cell differentiation and proliferation in response to GC stimulation [52].

Serum T-ALP is produced during the maturation of the bone matrix and is intimately associated with the mineralization of the bone matrix. It is one of the markers reflecting osteogenic activity in bone turnover [48,53]. ALP biochemical assays were used in our investigation to gauge osteoblast differentiation. ALP is a widely recognized biochemical indicator of osteoblasts due to the production of ALP enzymes during the differentiation process of osteoblasts [53]. As one of the phenotypic indicators of osteoblasts, bone ALP is important for osteogenesis. Its activity can be a direct indicator of osteoblast function or activity. DEX inhibits initial osteoblast proliferation, followed by decreasing ALP activity, causing a reduction in the ALP of the MC3T3-E1 cells [36]. Our study’s findings showed that the addition of PTT dose-dependently restored ALP activity reduced by DEX. The osteoblasts treated with DEX had a greater total ALP level on day six after being treated with PTT at varying concentrations of 0.25, 0.5, and 1 µg/mL. According to the findings, PTT supplementation increased ALP activity in a dose-dependent way. However, prolonged treatment of PTT did not show any positive changes. This could be due to the negative effects of DEX on osteoblast differentiation being too intense as the exposure was prolonged, and a higher dose of PTT might show some positive effects.

Alizarin Red stain (ARS) was used to detect calcium nodules (mineralized nodules), and cell mineralization was quantified by measuring the deposition. In this study, it was shown that DEX treatment caused a significant reduction in the calcium nodules in a dose- and time-dependent way. Even though differentiation into osteogenic lineage and mineralization can only occur in the presence of DEX, excessive and prolonged exposure to DEX will compromise this process [48]. Liu et al. 2018 also revealed similar findings where DEX caused decreased mineralization of the MC3T3-E1 cells [38]. With the addition of PTT, mineralization in MC3T3-E1 cells was found to be promoted, evidenced by the significantly higher calcium nodules found in the cells treated with 0.5 μg/mL of PTT on day 18. This effect could be due to PTT strengthening osteoblastic differentiation due to the enrichment of the pro-mineralization protein [33]. According to a study by Deng et al., mouse bone marrow cells treated with γ-tocotrienol formed more calcium nodules, and Wan Noraini et al. 2020 showed that annatto-derived tocotrienol exhibits an anabolic effect on osteoblast mineralization through inhibition of the mevalonate pathway, leading to activation RhoA, thus increases BMP-2 protein expression [33].

This study’s findings demonstrated that supplementing dexamethasone-treated cells with PTT increased the expression of Wnt3a, β-catenin, COL1α1, ALP, OCN, LRP6, and Runx2 protein. The significant increase occurred constantly in all proteins on day 6 and at a dose of 0.5 μg/mL PTT. The canonical Wnt signaling pathway is a key player in osteoblast differentiation, proliferation, and maturation. It also has a significant impact on controlling bone remodeling and development [54]. The expression of Runt-related transcription factor 2 (Runx2) and Osterix (SP7) transcription factors are activated by Wnt and BMP signaling, which is necessary for the differentiation of osteoblast progenitor cells into pre-osteoblasts and, subsequently, osteoblasts [12,13,55,56,57]. In the presence of a Wnt molecule, the Wnt/β-catenin signaling pathway becomes active. Osteoblasts and osteoblastic progenitor cells are directly impacted by β-catenin [58]. It controls the direction in which MSCs differentiate into osteoblasts and stimulates early osteoblast differentiation by increasing the response of these cells to BMP-2 via the Wnt/β-catenin pathway [59,60,61,62,63]. Wnt3a activates Runx2 through a synergistic interaction with BMP-9, promoting the production and deposition of bone matrix [64]. In order to activate tyrosine kinase (CKI) in cells and attract Disheveled (DVL) proteins to the cell membrane for phosphorylation, the Wnt ligand attaches to Frizzled (FRZ) and engages in interactions with the co-receptor LRP5/6. Osteoblasts express all elements of the canonical Wnt signaling pathway, including LRP5 and LRP6 [28,65,66]. Therefore, exposure to excess glucocorticoids suppresses Wnt signaling by reducing Wnt expression [67,68], enhancing the expression of Wnt antagonists [67,69], Sost [69,70,71], and Secreted frizzled-related protein-1 (sFRP-1) [67,72], and elevating the expression of Axin-2 [69], a negative Wnt signaling regulator. Notably, there is a decrease in the serum concentration of SOST in humans, which may indicate a compensatory mechanism that needs more research [73,74]. By increasing the expression of BMP antagonists and suppressing BMP-2 expression [68,73], glucocorticoids also reduce BMP signaling Da and Follistatin [72]. Furthermore, glucocorticoids prevent osteoblast maturation by suppressing Runx2 expression and Runx2 activity [8,74]. A study by Hong and Zhang 2020 found that hesperidin (HES), similar to PTT, which has anti-inflammatory, anti-oxidation activities, appears to regulate cell differentiation through the Wnt/β-catenin signaling pathway while also influencing the mineralization process by increasing the expression of osteogenic gene (ALP, OCN, OSX, and Runx2) in human alveolar osteoblasts [75].

As stated before, DEX suppresses osteoblast growth at first, which is followed by a reduction in ALP activity. A previous study also stated that expressions of the osteogenic genes (ALP, OCN, OSX, and Runx2) were downregulated due to DEX treatment [36]. This study showed that PTT supplementation at the dose of 0.5 μg/mL restored ALP activity and increased the expression of Wnt3a, CTNNB1, Runx2, COL1a1, and OCN. This finding was consistent with a study by Kim et al. (2021) that demonstrated how the antioxidant albiflorin increased the expression of genes related to bone formation, including OSN, OCN, and OPN, via the Wnt/β-catenin signaling pathway, which is beneficial for osteoblast differentiation in MC3T3-E1 cells [76].

The experimental results revealed that tocotrienol effectively promoted osteogenic differentiation in the MC3T3-E1 cells and improved osteogenesis, which was suppressed by DEX. Based on these findings, palm tocotrienol appears to be a promising candidate medication for the management of GIO. By employing tocotrienol, this research may offer a novel method for preventing osteoporosis brought on by GCs. Therefore, by controlling the Wnt3a-β catenin pathway, PTT may act as a possible prophylactic agent against GIO.

Like other similar studies, our study is not without technical limitations. One of the main limitations of this study is the employment of a single pre-osteoblast cell line, MC3T3-E1, which may not fully encapsulate the complex biological interactions in humans. Some cell death and marker determination results should be confirmed with additional experiments like apoptosis assay, Western blotting, or immunofluorescent analysis. Additionally, our study focuses on the potential osteogenic effects within a relatively short timeframe (up to 24 days). The long-term implications of DEX and PTT treatment on osteoblast viability and function remain inconclusive. Finally, although we highlight the involvement of the Wnt/β-catenin pathway, other molecular pathways influencing osteoblast differentiation and function remain unexplored and warrant further investigation.

## 5. Conclusions

In summary, our work showed that, in the presence of GC-induced dysfunction in MC3T3-E1 cells cultured in vitro, palm tocotrienol promoted osteogenic differentiation and bone production by preserving cell viability and stimulating osteogenic differentiation that DEX repressed. To completely understand the mechanism underlying GC-induced apoptosis in primary osteoblasts, more research is necessary. The findings of this study were consistent with our earlier research on animals, which may offer a unique method for treating GIO using tocotrienol in order to avoid osteoporosis caused by GCs.

## Figures and Tables

**Figure 1 biomedicines-13-00243-f001:**
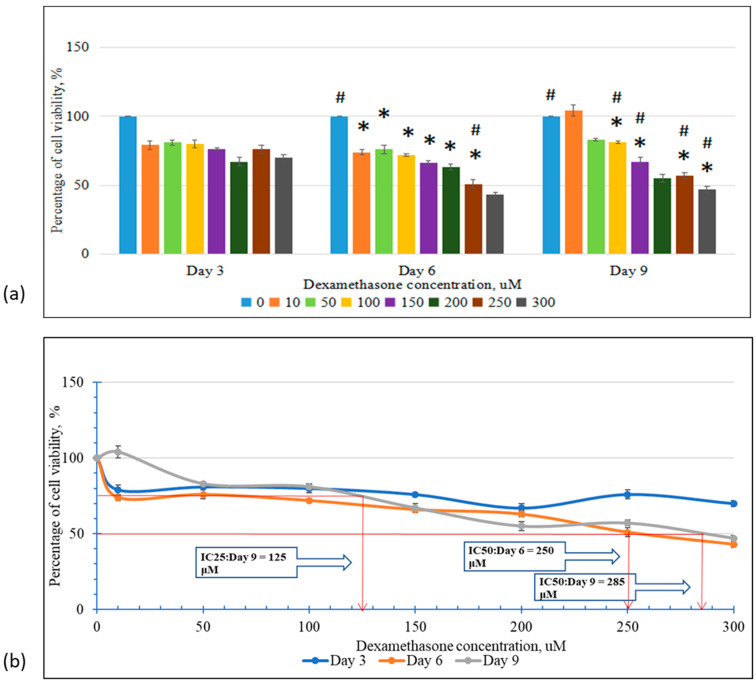
Cell viability (**a**) Effects of different concentrations of DEX on the viability of MC3T3-E cells for days 3, 6, and 9, (**b**) IC50 and IC25 for DEX treatments. Data are expressed as mean ± SEM (*n* = 3). # *p* < 0.05 compared to the control group. * *p* < 0.05 compared to the other treatment group on the same day. Data were analyzed using one-way ANOVA with Tukey’s post hoc analysis.

**Figure 2 biomedicines-13-00243-f002:**
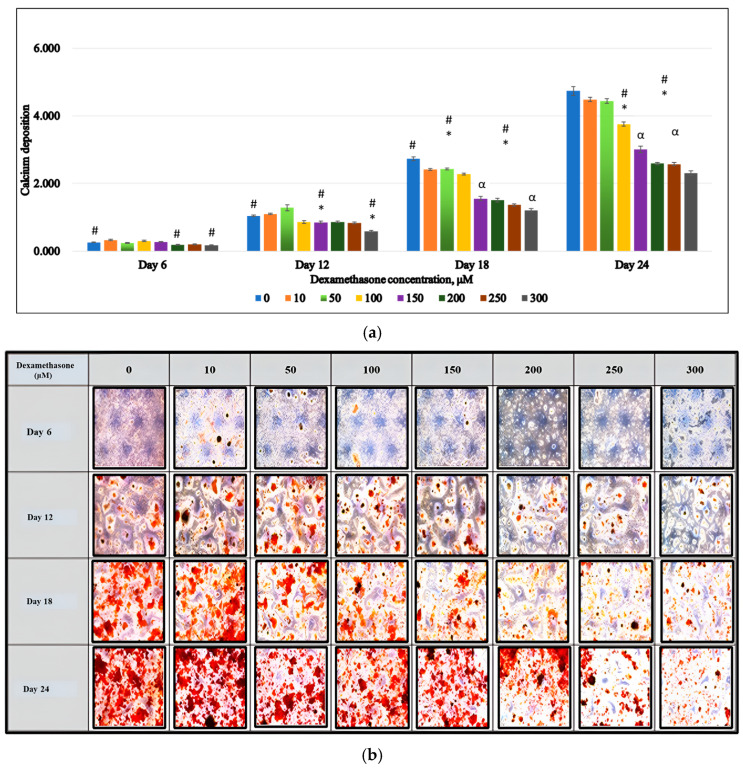
Effects of DEX on MC3T3-E cell differentiation. Cells treated with different doses of DEX (0–300 µM) on days 6, 12, 18, and 24. (**a**) Formation of mineralized nodule (100×), (**b**) Alizarin Red was quantified by spectrophotometer. The results are expressed as means ± SEM (n = 3). # *p* < 0.05, compared to the control group. * *p* < 0.05 compared to group D100 on the same day. α *p* < 0.05 compared to group D150 on the same day. Data were analyzed using one-way ANOVA with Tukey’s post hoc analysis.

**Figure 3 biomedicines-13-00243-f003:**
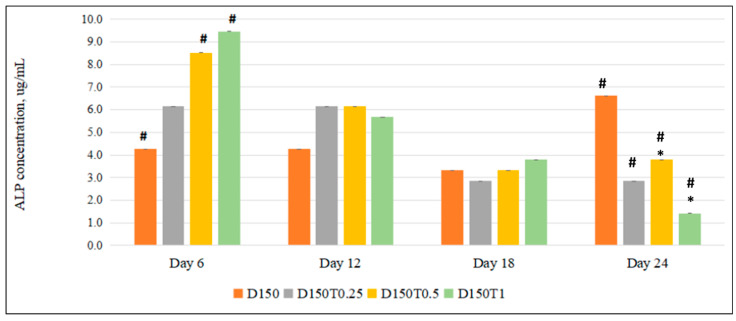
The effect of PPT on the ALP activity in DEX-treated MC3T3-E1 cells. MC3T3-E1 cells were treated with DEX and PPT for 6, 12, 18, and 24 days. Data are presented as the mean ± SEM. # *p* < 0.05 vs. control. D150: DEX 150 µM, D150T0.25: DEX 150 µM + PTT 0.25 µg/mL; D150T0.5: DEX 150 µM + PTT 0.5 µg/mL; D150T1: DEX 150 µM + PTT 1 µg/mL. # *p* < 0.05 compared to the control group. * *p* < 0.05 compared to the other treatment group on the same day. Data were analyzed using one-way ANOVA with Tukey’s post hoc analysis.

**Figure 4 biomedicines-13-00243-f004:**
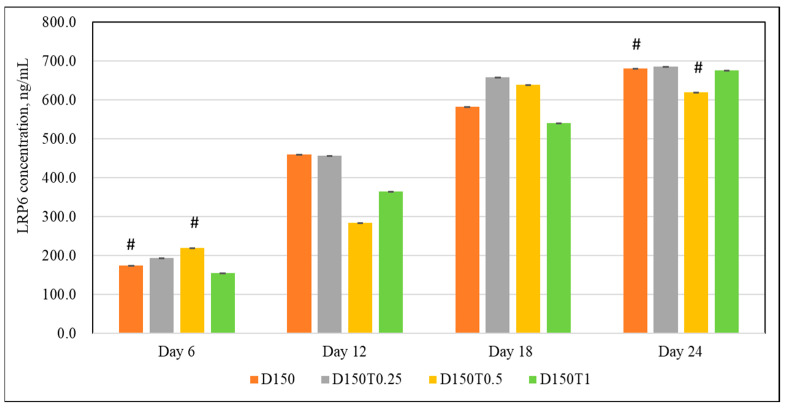
The effect of PPT on the LRP6 activity in DEX-treated MC3T3-E1 cells. MC3T3-E1 cells were treated with DEX and PPT for 6, 12, 18, and 24 days. Data are presented as the mean ± SEM. # *p* < 0.05 vs. control. D150: DEX 150 µM, D150T0.25: DEX 150 µM + PTT 0.25 µg/mL; D150T0.5: DEX 150 µM + PTT 0.5 µg/mL; D150T1: DEX 150 µM + PTT 1 µg/mL. # *p* < 0.05 compared to the control group. Data were analyzed using one-way ANOVA with Tukey’s post hoc analysis.

**Figure 5 biomedicines-13-00243-f005:**
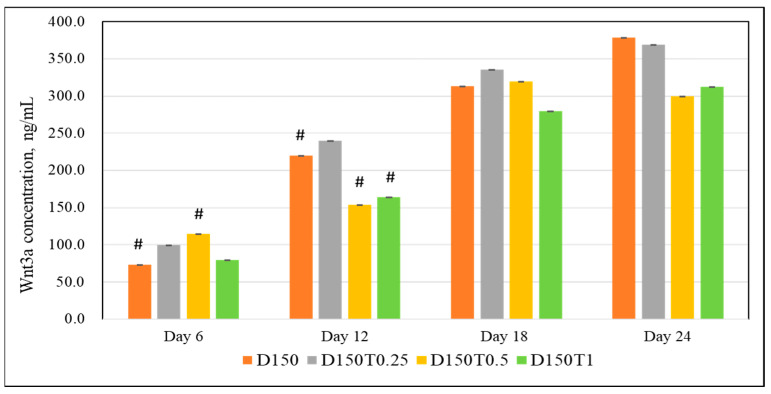
The effect of PTT on the Wnt3a activity in DEX-treated MC3T3-E1 cells. Data are presented as the mean ± SEM. # *p* < 0.05 vs. control. D150: DEX 150 µM, D150T0.25: DEX 150 µM + PTT 0.25 µg/mL; D150T0.5: DEX 150 µM + PTT 0.5 µg/mL; D150T1: DEX 150 µM + PTT 1 µg/mL. # *p* < 0.05 compared to the control group. Data were analyzed using one-way ANOVA with Tukey’s post hoc analysis.

**Figure 6 biomedicines-13-00243-f006:**
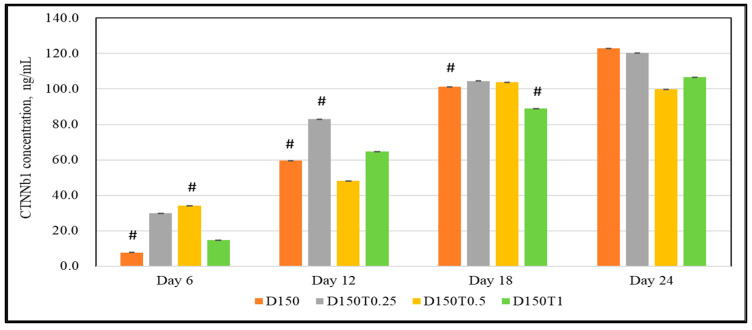
Effect of PPT on the CTNNb1 expression in DEX-treated MC3T3-E1 cells. Data are presented as the mean ± SEM. # *p* < 0.05 vs. control. D150: DEX 150 µM, D150T0.25: DEX 150 µM + PTT 0.25 µg/mL; D150T0.5: DEX 150 µM + PTT 0.5 µg/mL; D150T1: DEX 150 µM + PTT 1 µg/mL. # *p* < 0.05 compared to the control group. Data were analyzed using one-way ANOVA with Tukey’s post hoc analysis.

**Figure 7 biomedicines-13-00243-f007:**
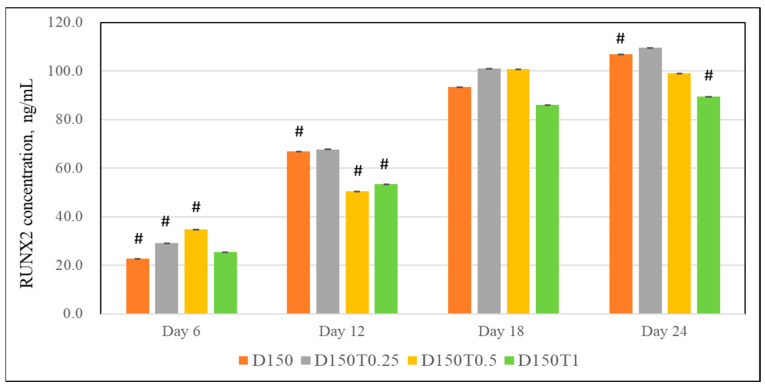
Effect of PPT on the RUNX2 expression in DEX-treated MC3T3-E1 cells. Data are presented as the mean ± SEM. # *p* < 0.05 vs. control. D150: DEX 150 µM, D150T0.25: DEX 150 µM + PTT 0.25 µg/mL; D150T0.5: DEX 150 µM + PTT 0.5 µg/mL; D150T1: DEX 150 µM + PTT 1 µg/mL. # *p* < 0.05 compared to the control group. Data were analyzed using one-way ANOVA with Tukey’s post hoc analysis.

**Figure 8 biomedicines-13-00243-f008:**
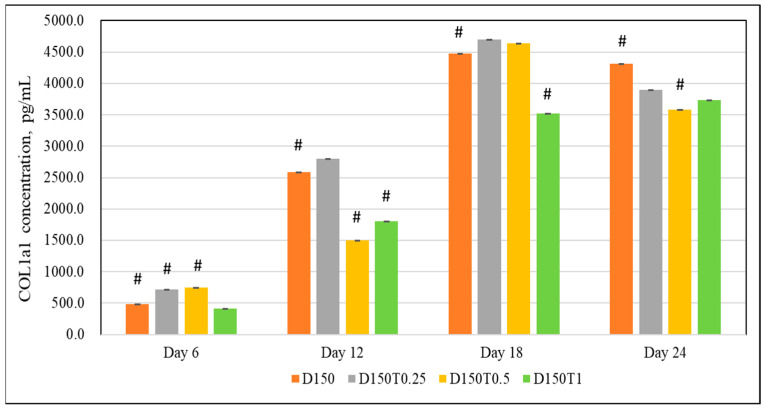
Effect of PPT in DEX-treated MC3T3-E1 cells. D: Dexamethasone, T: Tocotrienol. Data are presented as the mean ± SEM. # *p* < 0.05 vs. control. D150: DEX 150 µM, D150T0.25: DEX 150 µM + PTT 0.25 µg/mL; D150T0.5: DEX 150 µM + PTT 0.5 µg/mL; D150T1: DEX 150 µM + PTT 1 µg/mL. # *p* < 0.05 compared to the control group. Data were analyzed using one-way ANOVA with Tukey’s post hoc analysis.

**Figure 9 biomedicines-13-00243-f009:**
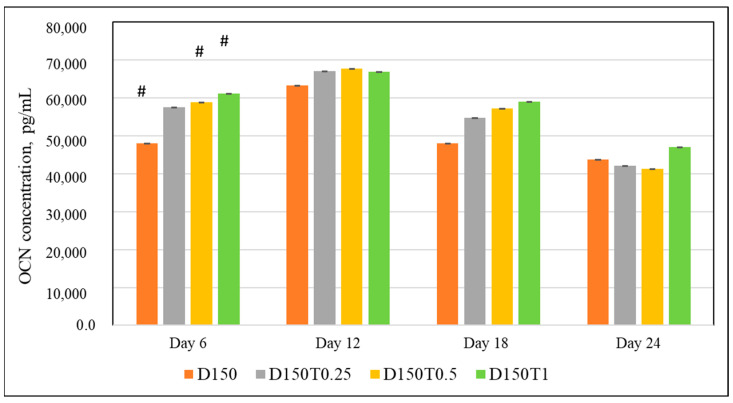
Effect of PPT in DEX-treated MC3T3-E1 cells. Data are presented as the mean ± SEM. # *p* < 0.05 vs. control. D150: DEX 150 µM, D150T0.25: DEX 150 µM + PTT 0.25 µg/mL; D150T0.5: DEX 150 µM + PTT 0.5 µg/mL; D150T1: DEX 150 µM + PTT 1 µg/mL. Data were analyzed using one-way ANOVA with Tukey’s post hoc analysis.

**Figure 10 biomedicines-13-00243-f010:**
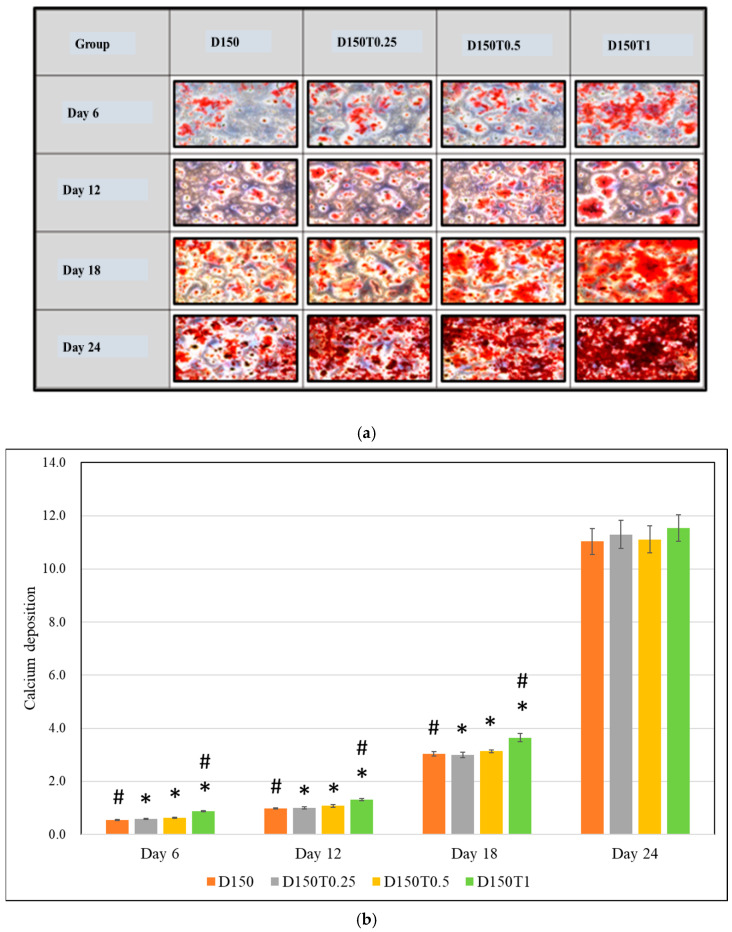
The effect of PPT on the mineralization of DEX-treated MC3T3-E1 cells. The mineralized matrix of MC3T3-E1 cells was stained with Alizarin Red S following incubation with osteogenic medium after day 6 to day 24. (**a**) microscopic examination of the samples. (**b**) Alizarin Red was quantified by spectrophotometer. The results are expressed as means ± SEM (*n* = 3). # *p* < 0.05, compared to the control group. D150: DEX 150 µM, D150T0.25: DEX 150 µM + PTT 0.25 µg/mL; D150T0.5: DEX 150 µM + PTT 0.5 µg/mL; D150T1: DEX 150 µM + PTT 1 µg/mL. * *p* < 0.05 compared to the other treatment group on the same day. Data were analyzed using one-way ANOVA with Tukey’s post hoc analysis.

## Data Availability

The original contributions presented in this study are included in the article. Further inquiries can be directed to the corresponding author.

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
