# Peer review of "Palm Tocotrienol Activates the Wnt3a/β-Catenin Signaling Pathway, Protecting MC3T3-E1 Osteoblasts from Cellular Damage Caused by Dexamethasone and Promoting Bone Formation"

_biomedicines, 2025, doi:10.3390/biomedicines13010243_

Round 1
Reviewer 1 Report
Comments and Suggestions for Authors
Manuscript biomedicines-3397208
Authors: Sani and his/her collaborators
In this research article entitled “Palm Tocotrienol Activates the Wnt3a/β-Catenin Signaling Pathway, Protecting MC3T3-E1 Osteoblasts from Cellular Damage Caused by Dexamethasone and Promoting Bone Formation”, the authors studied the mechanism of behind the effect of tocotrienol, originated from palm, on MC3T3-E1 murine preosteoblastic cells treated with glucocorticoid (GC). The authors reported that PTT maintains the osteogenic activity of the dexamethasone treated osteoblasts by promoting their differentiation, it may also act as a possible prophylactic agent against glucocorticoid-induced osteoporosis (GIO).
Hereafter, some points that should be taken into account before processing further.
Comments to the authors:
1- First of all, it is recommended to reduce the plagiarism percentage as it is 34%.
2- Regarding statistical analyses, some of the histograms are confusing because the figure bars showed possibilities of statistical differences but no symbols exist. For example, in figure 3, there would certainly be some significant statistical differences between the bars of “Day 24” but nothing is reported. The authors are requested to check it.
3- Quality of some figures should be ameliorated. Such as the boxes in figure 1b…
4- Some of the sentences “Bone undergoes a … relatively high turnover”, “GCs reduce … formation of new bone.”, and “The use of natural products … preventions in recent years.” can be supported by the following recent and relevant reference: Doi: 10.1016/j.vibspec.2021.103279.
5- Check for correct form of the unit “µM” in some figures.
6- It is recommended to add some limitations of the study, as this would have additional value to the manuscript, particularly its discussion.
7- English language and style are fine and just minor editing is required.
Author Response
Dear reviewer,
Thank you for your valuable comments and suggestions. We have done amendments to the manuscripts according to your suggestions as much as possible. Please find the attachment file.

Reviewer 2 Report
Comments and Suggestions for Authors
Title: Palm Tocotrienol Activates the Wnt3a/β-Catenin Signaling Pathway, Protecting MC3T3-E1 Osteoblasts from Cellular Damage Caused by Dexamethasone and Promoting Bone Formation
The major works in the submitted manuscript have been focused in the effects of PTT on osteogenic activity in the cell model. Overall, the manuscript has clearly presented the reasonable results and the methods used in the study are appropriate to the aims of the study. However, there are a number of issues with the presentation of results and analysis that need to be clarified and addressed as followings.
1. Palm oil derived-tocotrienol (PTT) was a gift from Excelvite
Please add the detailed data of the PTT compositions in the Methods.
2. A total of 45 µL of 1g/mL palm tocotrienol stock was activated by mixing it with….
The unclear meaning of “was activated” should be reconsidered by using the other sentence.
3. Figure 4. The effect of PPT on the on LRP6 activity in DEX-treated MC3T3-E1 cells
The sentence should be revised to adequate presentation.
4. Figure 10. # = p<0.05 compared to the control group. * = p<0.05 compared to the previous treatment group on the same day.
The “=” symbol should be deleted.
5. Figure 10. The absorbance was measured at 450nm
The absorbance value should be lower than 2.0 owing to the limit of the Beer-Lambert Law.
Comments on the Quality of English Language
There was no comment on the English quality.
Author Response

(The authors gave the same response as above.)

Reviewer 3 Report
Comments and Suggestions for Authors
This manuscript investigates the effects of palm oil-derived tocotrienol (PTT) on MC3T3-E1 osteoblasts, particularly its ability to improve cell viability and mineralization under dexamethasone-induced cellular damage. The results suggest that PTT, at certain concentrations, promotes osteoblast mineralization and enhances cell growth and bone formation capacity. While the study presents meaningful findings and the experimental design appears reasonable, several key issues need to be addressed.
1. The study suggests that PTT exerts its effects through the Wnt3a/β-catenin pathway, but the current mechanistic evidence is limited. Additional experiments, such as employing Wnt inhibitors or β-catenin knockdown models, are recommended to robustly validate the pathway's role.
2. The use of ELISA to measure Wnt3a, β-catenin, COL1α1, ALP, OCN, LRP6, and RUNX2 raises concerns. While appropriate for secreted proteins like Wnt3a, ALP, and OCN, it is unclear how intracellular or membrane-bound proteins (e.g., β-catenin, RUNX2, LRP6) were detected in the supernatant. The lack of clarification on sample preparation and ELISA kit specificity undermines the validity of the method. The authors should explain whether cell lysates were used and verify intracellular protein levels through complementary methods like Western blot or immunofluorescence.
3. The manuscript primarily uses MTT assays to evaluate the protective effects of PTT on cell viability, but this method has limitations and does not fully reflect actual cell survival. The manuscript claims that "cell viability significantly rebounded," but the MTT results show only a modest improvement (10-20%). This limited improvement is insufficient to support such a conclusion. I recommend adding more assays, such as apoptosis assays (e.g., Annexin V staining or TUNEL assay), to provide stronger evidence.
4. The statistical annotations in all figures are incomplete, which undermines the clarity and reliability of the results. Additionally, the explanation of statistical markers in the figure legends is unclear and difficult to interpret. The authors should provide complete and clearly defined statistical annotations for all figures and ensure that the legends explain the markers in an easily understandable manner.
5. The manuscript contains multiple instances of "(Error! Reference source not found)" in citations. This oversight affects the readability and reliability of the work and should be corrected.
6. The authors should indicate the number of replicates/animals and the type of statistical analysis used for each figure and study.
7. Some recent works on how natural products improve the osteogenesis of bone cells are missing, such as Li W et al, Journal of dental research. 2022;101(7):802-811. Wu Y et al, International Journal of Molecular Sciences. 2024,25(5):2947. Wang X et al, Materials Toady Chemistry. 2024, 38:102052. Liu Z, Journal of Periodontal Research. 2023,58(6):1300-1314.
Author Response

(The authors gave the same response as above.)

Round 2
Reviewer 3 Report
Comments and Suggestions for Authors
Please send a clean version.
Author Response
Dear editor and reviewer
Thank you for highlighting the relevant issues. We are sorry for the errors.
(I) Ensure all references are relevant to the content of the manuscript.
The list and references have been updated
(II) Highlight any revisions to the manuscript, so editors and reviewers can
see any changes made.
We are sorry for the trouble. The revised sentences have been written in red and the rephrased paragraph has been highlighted in yellow
(III) Provide a cover letter to respond to the reviewers’ comments and
explain, point by point, the details of the manuscript revisions.
The cover letter and list of changes were included.
(IV) If the reviewer(s) recommended references, critically analyze them to
ensure that their inclusion would enhance your manuscript. If you believe
these references are unnecessary, you should not include them.
Some new references have been included.
(V) If you found it impossible to address certain comments in the review
reports, include an explanation in your appeal

Round 3
Reviewer 3 Report
Comments and Suggestions for Authors
Dear authors
Please send a clean and final version.
Currently, I can find no substantial changes but only your own comments in track mode.
Thanks
Author Response
Dear editor and reviewers,
I am resubmitting the revised clean version of the manuscript of our original article entitled ‘Palm Tocotrienol Activates the Wnt3a/β-Catenin Signaling Pathway, Protecting MC3T3-E1 Osteoblasts from Cellular Damage Caused by Dexamethasone and Promoting Bone Formation for consideration in your journal. We really appreciate the comments and suggestions. We have edited the manuscript accordingly as suggested by the reviewers to the best that we could and we hope it fulfils the requirements. We tried our best to respond to the reviewers’ comments and suggestions and we hope the latest version of the manuscript meets you journal standard. We are really sorry for inconvenience caused by uploading the previous version of the manuscript.
For the latest version, the changes were written in red.
I really appreciate your consideration in accepting and publishing this manuscript and we are willing to fulfil further requirements and changes or corrections to be done in order suit the standard of your journal.
The comments from the reviewer:
Please send the clean version of the manuscript.
Response: We have uploaded the clean version of the manuscript.
We are sorry for the trouble. The revised sentences from round 1 and 2 revisions have been written in red and the rephrased paragraph has been highlighted in yellow.
Yours sincerely,
Elvy Suhana Mohd Ramli
(Corresponding author).

Round 4
Reviewer 3 Report
Comments and Suggestions for Authors
Congrats, I have no more comments.